The evolution of SARS-CoV-2 and the COVID-19 pandemic

Si Yuanfang 1 2 3
Wu Weidong 4
Xue Xia 1 2 3
Sun Xiangdong 1 2 3
Qin Yaping 5
Li Ya 1 2 3
Qiu Chunjing 1 2 3
Li Yingying 1 2 3
Zhuo Ziran 4
Mi Yang 1 2 3 yangmi198@zzu.edu.cn
Zheng Pengyuan pyzheng@zzu.edu.cn 1 2 3
1 Henan Key Laboratory of Helicobacter Pylori & Microbiota and Gastrointestinal Cancer, Marshall Medical Research Cente, The Fifth Affiliated Hospital of Zhengzhou University , Zhengzhou , Henan , China
2 Department of Gastroenterology, The Fifth Affiliated Hospital of Zhengzhou University , Zhengzhou , Henan , China
3 Academy of Medical Science, Zhengzhou University , Zhengzhou , Henan , China
4 BGI College & Henan Institute of Medical and Pharmaceutical Sciences, Zhengzhou University , Zhengzhou , Henan , China
5 School of Basic Medical Sciences, Henan Institute of Medical and Pharmaceutical Sciences, Zhengzhou University , Zhengzhou , Henan , China
Pazzaglia Mariella
Electronic publication date: 2023 Sep 7
Publication date: 2023
Volume: 11
Electronic Location ID: e15990
Received 2023 Mar 21; Accepted 2023 Aug 8
Copyright: ©2023 Si et al.
Copyright year: 2023
Copyright holder: Si et al.
License: This is an open access article distributed under the terms of the Creative Commons Attribution License, which permits unrestricted use, distribution, reproduction and adaptation in any medium and for any purpose provided that it is properly attributed. For attribution, the original author(s), title, publication source (PeerJ) and either DOI or URL of the article must be cited.
License URL: https://creativecommons.org/licenses/by/4.0/

Keywords: Evolution, Populations, Interactions, Epidemiology, Viral Proteins

Funding: Zhengzhou Major Collaborative Innovation Project 18XTZX12003 Key projects of discipline construction in Zhengzhou University XKZDJC202001 National key Research and development program in China 2020YFC2006100 This work was supported by the Zhengzhou Major Collaborative Innovation Project (No. 18XTZX12003), the Key projects of discipline construction in Zhengzhou University (No. XKZDJC202001), and the National key Research and development program in China (No. 2020YFC2006100). The funders had no role in study design, data collection and analysis, decision to publish, or preparation of the manuscript.

==============================
Scientists have made great efforts to understand the evolution of SARS-CoV-2 (Severe Acute Respiratory Syndrome Coronavirus 2) to provide crucial information to public health experts on strategies to control this viral pathogen. The pandemic of the coronavirus disease that began in 2019, COVID-19, lasted nearly three years, and nearly all countries have set different epidemic prevention policies for this virus. The continuous evolution of SARS-CoV-2 alters its pathogenicity and infectivity in human hosts, thus the policy and treatments have been continually adjusted. Based on our previous study on the dynamics of binding ability prediction between the COVID-19 spike protein and human ACE2, the present study mined over 10 million sequences and epidemiological data of SARS-CoV-2 during 2020-2022 to understand the evolutionary path of SARS-CoV-2. We analyzed and predicted the mutation rates of the whole genome and main proteins of SARS-CoV-2 from different populations to understand the adaptive relationship between humans and COVID-19. Our study identified a correlation of the mutation rates from each protein of SARS-CoV-2 and various human populations. Overall, this analysis provides a scientific basis for developing data-driven strategies to confront human pathogens.

Introduction

The disastrous effects of large-scale infectious diseases have never been far from humans (Taylor, 2022), including the latest pandemic of COVID-19 (Coronavirus disease 2019) that has caused countless deaths (Baig et al., 2021). COVID-19 is caused by SARS-CoV-2 (Severe Acute Respiratory Syndrome Coronavirus 2), a novel coronavirus that first emerged in Wuhan, China, in December 2019 (Sharma, Kaur & Narwal, 2020), which is a respiratory virus primarily transmitted through respiratory droplets (Zhou et al., 2021). As the COVID-19 outbreak has been worldwide for over three years, our understanding of SARS-CoV-2 overall has enhanced, and it has been confirmed that the clinical and genetic information on this disease is particularly valuable for the global public health (Rahman et al., 2022). Scientists have fought SARS-CoV-2 by tracking its epidemiology, physical protection, developing vaccines, and effective treatments (Maduray & Parboosing, 2021). However, as an RNA (ribonucleic acid) viral pathogen, it is challenging to predict and prevent the mutation of SARS-CoV-2 due to its unique biological signatures (Albery et al., 2021).

The mRNA (messenger ribonucleic acid) polymerase error rate and recombination of viruses allow them to break species barriers and broaden their host range (Minardi da Cruz et al., 2013; Sanjuán & Domingo-Calap, 2021). Due to the lack of the proofreading function, a high error rate from RNA replicative enzymes brings a higher mutation rate in RNA virus, which is around 10−6 to 10−4 substitutions per nucleotide site per cell infection (s/n/c) (Peck & Lauring, 2018). Thus, the interaction between hosts and RNA pathogens is dramatically complicated, particularly in terms of their co-evolution (Jennings & Sang, 2019).

In this case, the co-evolution of humans and SARS-CoV-2 is an ongoing process that has shaped the immune system of humans and the biological characteristics of pathogens (Sattenspiel, 2015). Various infectious diseases have emerged throughout history and wreaked havoc on human populations (Piret & Boivin, 2020). Pathogens have evolved strategies to survive and thrive in human populations, including rapidly mutating to evade the immune system and manipulating human behavior for efficient transmission in a large population (Seal, Dharmarajan & Khan, 2021). Human populations have also evolved in response to pathogen pressure, developing genetic adaptations that confer resistance or partial immunity (Liston et al., 2021). Regardless of artificial interference, new viral pathogens initiating an outbreak in any new host would eventually evolve so that they could spread as widely as possible without eliminating the natural population of their host (Dennehy, 2017). However, the evolutionary trade-off of viral pathogenicity is difficult to measure or evaluate due to their high transmissivity and virulence, while their biological aim is relatively straightforward, that is, replicating and dispensing genetic materials, including DNA (deoxyribonucleic acid) or RNA, in large quantities, and as fast as possible (Barr & Fearns, 2016). Therefore, understanding the evolutionary mechanism behind this unpredictable behavior of viral pathogens would provide key strategies for COVID-19 and other viral pathogens monitoring and treatment (Luo et al., 2021).

The evolutionary origin of SARS-CoV-2 remains controversial, however, scientists believe that the virus possibly originated in bats before being transmitted to an intermediate host, likely a pangolin, where it mutated and then transferred to humans (Boni et al., 2020). Most viral evolution occurs by the gradual, stochastic accumulation of substitutions over lineage generations in adapting host environments that might ignore population heterogeneity (Day et al., 2020). Alternatively, viral evolution can be caused by a cluster of mutations that appear simultaneously, resulting in a sudden jump in viral evolution (Lauring, Frydman & Andino, 2013). In the COVID-19 pandemic, variants of concern (VOC), variations in Alpha, Beta, Gamma, Delta, and Omicron show extensive genomic mutations from the original Wuhan-01 strain, with enhanced viral infectivity and immune jeopardizing in humans (Nasir et al., 2022; Vangeel et al., 2022). To date, Omicron is the most divergent VOC (or strain) that spreads unpredictably fast and crafty, which has completed the evolutionary jump by infecting a large population of human hosts as the Delta outbreak in India in 2021 (Manjunath et al., 2022; Farahat et al., 2022; Farahat, Baklola & Umar, 2022). Although viruses constantly evolve and mutate, it is worth noting that the mutations do not necessarily mean that a virus becomes more dangerous or transmissible. Therefore, it is valuable to understand the mutation or evolutionary pattern based on big data of SARS-CoV-2 for further developments of vaccines and treatments for COVID-19.

This big-data analysis use of bioinformatics mining aims to reveal the evolutionary path of SARS-CoV-2 in various populations from different countries. We investigated 13,396,972 genomic sequences of SARS-CoV-2 from GISAID (https://gisaid.org/hcov19-variants/) and calculated the correlation of its mutation rate, COVID-19 cases, and death. Furthermore, based on the mutation rate of the main structure proteins (N, nucleocapsid protein; M, membrane protein; S, spike protein; E, envelop protein) in SARS-CoV-2, we determined the correlation between the mutation rates of SARS-CoV-2 and different populations.

Materials & Methods

Experimental design

Our study investigated the sequences of main proteins (N, M, S, and E) of SARS-Cov2 based on their genomics collected from different regions globally. We calculated and visualized the mutation rates of these proteins from different populations and COVID-19 timelines with the in-house bioinformatic framework. Furthermore, the reported COVID-19 cases from different regions were considered and included in our correlation analysis.

All data analysis and visualization via Python (v3.8.16) and the datasets were processed on a Linux-based DELL Precision 3640 workstation [Intel (R) Core (TM) i9-10900K CPU @ 3.70 GHz; 20 total cores enabled; 64 GB RAM].

In-house scripts were supplied in GitHub: https://github.com/wvdon/rate4sars.

Data collection

In this analysis, 13,396,972 genomic sequences of SARS-CoV-2 (1,918 lineages) were obtained from GISAID (https://gisaid.org/hcov19-variants/) between 2019-12-30 to 2022-10-09. In parallel, the additional information, including the accession ID, collection date, location, and Pango lineage, were included in our prediction. The reported COVID-19 cases were collected from COVID-19 Infection Data Realtime Crawler (https://github.com/BlankerL/DXY-COVID-19-Data) between 2020-01-24 to 2022-09-28.

Data processing

A global alignment of hCoV-19 data was obtained from GISAID and aligned to the reference sequence WIV04 of SARS-CoV-2 (Zhou et al., 2020). Based on the GISAID alignment results, the 13,396,972 sequences data were included in our analysis with 1,918 lineages.

Lineage date. To ensure the exact date of emergence of each lineage, we designed the following strategy:

(1) The emergence date of each lineage was confirmed as the date as the Pangolin website (https://cov-lineages.org/lineage_list.html) contains the date of emergence of the lineage.

(2) If there is not, we confirmed the date from the collection date of the lineage.

(a) At first, we filtered the collection date before Lineage A (2019-12-30), which was only in determining the date of lineages.

(b) Then, the earliest collection date was counted as the date of the lineage.

The table of the lineages we used in this work was shown in final_lineage_date.csv (GitHub repository: https://github.com/wvdon/rate4sars).

Sequence regions. To distinguish the data from different regions, the location from the metadata that matched it by keywords for different areas (as continents) and countries were considered. We filtered the sequences that missed the location information and plotted different areas map via Pygal (v3.0.0).

Lineage origin. In order to determine the evolutionary origin of each lineage, we used the Pango Nomenclature (Rambaut et al., 2020), which shows the relationship between the evolution of variants.

(1) To facilitate the calculation of the origin of the Pango lineage, we decompressed the alias key to full lineage name, such as P.1 to a full lineage name B.1.1.28.1.

(2) So, the evolutionary origin of P.1 (B.1.1.28.1) can be counted as B.1.1.28.

All the alias keys were obtained from cov-lineages (O’Toole et al., 2021).

Data analysis

The mutation rate of each protein was calculated by comparison with their parent clusters using the formula below (Lynch et al., 2016).

(1) pi=PBx−PBy

(2) Hp1,…,p22=−∑i=122pi⋅ log2pi

(3) Kaa=∑j=1NHNaa2T.

We calculated the mutation rates of N, M, S, and E proteins of SARS-CoV-2 that represents their fitness in human hosts, moreover, conducted correlation analysis in between.

Four structural proteins must ensure precise assembly into a functional virion to assemble an active virus. The mutation rate of SARS-CoV-2 from different areas is shown in Fig. 1. The figures were constructed by Matplotlib (v3.5.2; https://matplotlib.org/) and Seaborn (v0.11.2; https://seaborn.pydata.org/whatsnew/v0.11.2.html). The correlation analysis was conducted in Pandas (v1.4.3; https://pandas.pydata.org/docs/whatsnew/v1.4.3.html) with the Pearson algorithm (The Pandas Development Team, 2020), a widely used correlation statistics to measure the degree of relationship between linearly correlated variables more appropriate to our data. The parameters were set as default.

Figure 1 The sequence number and infection data of different areas or countries.

(A) Visualization of infection data in six continents and five countries. (B & C) The number of sequences from six continents and five countries.

The heatmap in this work was generated using Seaborn (v0.11.2). Dimensionality reduction comes under the heading of unsupervised machine learning algorithms meaning. This method aims to reduce the number of features in a dataset to aid in visualizing the mutation rate in different human populations after our analysis. PCA (Principal Component Analysis) was conducted to extract a new set of variables from a large set of ones, with these new variables taking the form of principal components, t-SNE was also performed to reduce the dimensionality in our correlation analysis. PCA accounts for non-linear relationships, which retains both structures in the lower dimension data by calculating the probability similarity of points in high and low dimensional space. Therefore, we visualized the distribution of mutation rate in different areas and countries by reducing the dimensionality to 2 dimensions. T-SNE and PCA visualizing analysis was completed by using Scikit-Learn (v1.0.2; http://scikit-learn.org/stable/whats_new/v1.0.html).

Results

We collected sequences and infection data from six continents and five countries, which was marked on the world map (Fig. 1A). It was found that the most significant number of COVID-19 sequences was from Europe as a continent, while Britain showed the largest as a country (Figs. 1B and 1C). We included 2,621 sequences from China, the smallest part of this analysis. According to the statistics, the total sequence number of each continent is 13,074,212. The location attribute of 322,760 was missing and cannot be counted by location. This work includes all genomic sequences of SARS-Cov-2 and COVID-19 cases from different regions and builds a correlation model.

Based on genomic and S protein mutation rates from 13,396,972 sequences of SARS-CoV-2, we revealed the relationship between the mutation rate of S protein in SARS-CoV-2 and the earliest emergence time of each lineage (1,918 in total). During the COVID-19 pandemic, the highest mutation rate of S protein showed around April 2020, which was consistent with Delta occurrence in Asia and North America (Fig. 2). The Pango counts refers to the number of new variations at that time, which counts the number of new variations by day. We found that number and frequency of the Pango counts and mutation were reduced along with the COVID-19 pandemic around the world, which suggests S protein might play a role in determining the antigenic evolution and increasing the transmissibility of SARS-CoV-2.

Figure 2 The relationship between the mutation rate of S protein from six continents and concurrence time, and the pango counts refers to the number of new lineage at the occurrence time points, which counts the number of new variations by day.

In addition, we calculated the mutation rates of N, M, S, and E proteins of SARS-CoV-2. Although it was confirmed that four proteins attach and work closely related to assemble a functional virus, it is hard to determine the evolutionary connection between each protein. Notably, we found that the S protein was less correlated with the M protein, indicating that the functions of infecting and assembling occur not simultaneously by activating various genetic pathways in the interactive system of the human virus (Fig. 3).

Figure 3 Correlation of mutation rates in N, S, E, and M proteins of SARS-CoV-2.

Furthermore, we investigated the substitution rate of N, M, S, and E proteins of SARS-CoV-2 between 2019-12-30 to 2022-10-09 and the fatality of COVID-19 between 2020-01-24 to 2022-09-28 (Fig. 4). Similar to the mutation rate of S protein in SARS-CoV-2, the death rate caused by COVID-19 showed highest around May in 2020 and went lower afterward. In parallel, COVID-19 cases increased continuously from 2020 to 2022. Furthermore, the mutation rate of N, M, S, and E proteins was similar to S protein, except a death peak showed at the beginning of 2021, which was later that the records of another VOC, Delta, emerged. Unlike the deaths caused by the earlier VOC, the date Omicron occurred performed higher infectious than death numbers. Notably, the number of death cases increased dramatically from 252,514 to 2,544,455 after the correction on November 2020. Our analysis potentially indicates the adaptation direction of SARS-CoV-2 towards high infection in human-to-human transmission under a selective environment.

Figure 4 The evolutionary events of SARS-CoV-2 and global public health.

The lineages that were larger than 0.02 were marked. The origin was the total number of confirmed cases in each country daily. A correction was that it counts after the day with incomplete data (not all countries reported cases every day). Dead Rate True was the original ratio, and the Correction Dead Rate was after removing the outliers.

To determine various epidemic mutants of SARS-CoV-2 escape from distinct human hosts with diverse ethnic and genetic backgrounds, the correlation between SARS-CoV-2 and cases from different regions was investigated. Based on the analysis of regions and viral variations, no clear patterns of its genomic changing or shift were identified in different human populations worldwide (Fig. 5), even though specific human genetic variants have been strongly associated with COVID-19 transmission.

Figure 5 The mutation rate visualization after dimensionality reduction.

(A) t-distributed stochastic neighbourhood embedding (t-SNE) dimensionality reduction of different Area or Country. (B) Principal component analysis (PCA) dimensionality reduction of different area or country.

Discussion

Since the World Health Organization (WHO) confirmed and announced the COVID-19 pandemic in 2020 (March), accumulating genomic sequences have been reported, and new lineage has been shown beyond expected. The present analysis aims to find out the mutation pattern of the main proteins of SARS-CoV-2 globally at different time stages of COVID-19. We included the genomic sequences of SARS-CoV-2 between 2020 to 2022 from different continents and countries (Fig. 1). We found that the deaths showed high around mid-2020 (Fig. 4) that partially due to the limited understanding of this pathogen and the lack of health care during the quick and server pandemic (Fajar et al., 2022; Lee & Neimeyer, 2022). We previously found the co-evolution between SARS-CoV-2 and human ACE2 based on 2,092 genome sequences of SARS-CoV-2 from over 50 countries. Moreover, it revealed 3,860 amino acid mutations in its spike protein RBD (T333-C525), which can set the complex with human ACE2 (Xue et al., 2021). Based on evolutionary progression in human hosts, two types of variants that we predicted have been identified in SARS-CoV-2 (Day et al., 2020; Luo et al., 2021).

Several studies have suggested that the long-term evolutionary adaptation of SARS-CoV-2 and limited genomic surveillance are the reasons for this viral evolution (Fig. 2). Since the S protein of SARS-CoV-2 plays a crucial role in viral entry to host cells by recognizing receptors and inducing cell membrane fusion, the bulk of studies thus far have focused on the spike protein (Singh et al., 2021; Palladino et al., 2022). It has been shown that spike protein determines the antigenic evolution and increasing transmissibility of SARS-CoV-2 (Lopez-Cortes et al., 2021; Magazine et al., 2022). Moreover, increasing evidence has found other SARS-CoV-2 proteins contributing more to virus-host interaction than expected (Takano et al., 2023). The cases of COVID-19 death nowadays kept low due to the highly improved public health readiness and more appropriate medical care.

Unlike the beginning stage of COVID-19, mRNA and adenovirus vector-based vaccines have been widely applied globally with increasing coverage. In China, over 70% of the population has completed the vaccination (Fajar et al., 2022). However, the strong spread rate and fast antigenic drifts in new lineages, such as Omicron, attenuate the power of vaccines and acquired immunity in previously infected individuals (Karim & Karim, 2021; Daria & Islam, 2022). Phylogenetic analysis during this pandemic has found that both steady accumulations of substitutions and changes of VOC occur in SARS-CoV-2 along with its transmission and infection (Fig. 4). The immunity acquired by previous infection or vaccination in humans has been found to be effective in preventing severe COVID-19 symposiums and reducing the hospitalization rate (Fajar et al., 2022). Although the apparent much-reduced severity to public health is not necessarily a direct manifestation of the decreasing virulence of SARS-CoV-2 with constant evolution through the pandemic (Rahman et al., 2022), there is a trade-off between transmissibility and virulence for SARS-CoV-2 as the dominant lineage, Omicron, showed relatively lower virulence compared with other VOC while its transmissibility increased over 100% (Karim & Karim, 2021; Vangeel et al., 2022). Moreover, the infection population size provides an advantageous evolutionary resource (genetic resources) for SARS-CoV-2 (Day et al., 2020). The Alpha/Beta in the UK and the Delta in India speed up the mutation rate of SARS-CoV-2 through a fast local outbreak within a certain geological and ethnic population (Karim & Karim, 2021; Daria & Islam, 2022). In addition, both vaccination and infection can change the immune response in the human population, enhance or constrain the adaptation direction of SARS-CoV-2, and somehow improve the virus find a way to interact with a human being more wisely, such as a long incubation period and immune escape (Fajar et al., 2022).

Different infection periods within a population are often caused by different variants of the explosive epidemic or powerful large-scale epidemics (Wilder-Smith, 2021; Weiss & Sankaran, 2022). The VOC that emerged during repeated outbreaks show a certain degree of immune escape against vaccines designed for the variants from the previous outbreaks (Nasir et al., 2022). The effect of vaccination in an extensive range is greatly reduced due to the constant variation of the virus (Puhach et al., 2022). The adaptive mutation or evolution of the virus poses a challenge to preventing and controlling this viral pathogen. Although we can predict the variation of the S protein of SARS-CoV-2, particularly in binding regions to hACE2, we know little about the evolutionary pathway of its overall mutation on the genome for different mammalian hosts, especially humans. Omicron may originate from mice, and an emerging virus in a human was confirmed from white-tailed deer (Du, Gao & Wang, 2022). As the SARS-CoV-1 pandemic found, SARS-CoV-2 highly likely originated from horseshoe bats and obtained human ACE2 binding ability by recombination in the intermediate reservoir (Tang et al., 2022), it evolves fast and unpredictable successfully in human beings and other animals, such as cats, civets, raccoons, and rodents (Ye et al., 2020). Due to the lack of investigation into the transmission of SARS-CoV-2, it is difficult to establish an epidemiological link among different hosts and to predict the possibility of reverse zoonosis (Tang et al., 2022). Through the detection of virus mutation, it can be found that multiple sites of its genome have undergone continuous mutation (Figs. 2 & 3). The domestic and international research currently focuses on virus spike proteins related to virus-infected cells and binding to human ACE2 receptors (primarily its RBD variable domain) (Lopez-Cortes et al., 2021; Magazine et al., 2022). Although this analysis includes as much information as SARS-CoV-2 genomics and COVID-19 cases, the missing data from specific regions limits the explanatory power of our calculated pattern. In addition, the sequencing bias from datasets without clinical information makes it challenging to confirm our model precisely in the real world.

Therefore, the precise coordination mechanism of virus evolution with its four structural proteins by virus packaging as well as the understanding of virus immune escape strategies other than the S protein is still in a relatively shallow stage, which severely limits our scientific foresight on the variation of COVID-19 in different populations, and further hinders the development of the targeted and prospective prevention, control, and treatment methods based on specific mutation biological significance.

Conclusions

This analysis shows that SARS-CoV-2 does not mean killing the host in large numbers and quickly. We confirmed that there is no population preference for SARS-CoV-2 infection. Nowadays, most countries have announced that the COVID-19 pandemic is“over” and vaccines prevent severe life-threatening conditions for most people, however, preventing or treating this disease is not generally accomplished. The ultimate intervention strategy for terminating this pandemic is to deploy general and efficient vaccination against SARS-CoV-2 across all age and ethnic groups. It is still challenging due to the high mutation rate and strong adaptation of SARS-CoV-2 that enhances its host immune evasiveness and host-to-host transmissibility, especially in countries with low vaccination rates. Our analysis thus provides a genetic understanding of SARS-CoV-2 from different populations and sets a foundation to discover efficient strategies for preventing and treating this viral pathogen.

Additional Information and Declarations

Competing Interests

Author Contributions

Data Availability

The authors declare there are no competing interests.

Yuanfang Si performed the experiments, authored or reviewed drafts of the article, and approved the final draft.

Weidong Wu conceived and designed the experiments, performed the experiments, authored or reviewed drafts of the article, and approved the final draft.

Xia Xue conceived and designed the experiments, performed the experiments, authored or reviewed drafts of the article, and approved the final draft.

Xiangdong Sun analyzed the data, prepared figures and/or tables, and approved the final draft.

Yaping Qin analyzed the data, prepared figures and/or tables, and approved the final draft.

Ya Li analyzed the data, prepared figures and/or tables, and approved the final draft.

Chunjing Qiu analyzed the data, prepared figures and/or tables, and approved the final draft.

Yingying Li analyzed the data, prepared figures and/or tables, and approved the final draft.

Ziran Zhuo performed the experiments, analyzed the data, prepared figures and/or tables, and approved the final draft.

Yang Mi conceived and designed the experiments, authored or reviewed drafts of the article, and approved the final draft.

Pengyuan Zheng conceived and designed the experiments, authored or reviewed drafts of the article, and approved the final draft.

The following information was supplied regarding data availability:

The code and fig are available at GitHub and Zenodo:

- https://github.com/wvdon/rate4sars.

- wvdon. (2023). wvdon/rate4sars: version_1.0.0_may_26 (version1.0.0). Zenodo. https://doi.org/10.5281/zenodo.7972733.

The findings of this study are based on metadata associated with 13,396,972 sequences available on GISAID up to October 9, 2022, and are available at https://doi.org/10.55876/gis8.220330me.

The reported COVID-19 Infection Data is available at GitHub:

https://github.com/BlankerL/DXY-COVID-19-Data/releases.

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
