# Peer review of "The evolution of SARS-CoV-2 and the COVID-19 pandemic"

_PeerJ, doi:10.7717/peerj.15990_

## Round 0.1 · original submission · Major Revisions

As usual, I have invited comments from experts from your research domain.

As you will see, multiple limitations were highlighted by reviewers that would need to be addressed/explained carefully.

Taken altogether, let me invite you to prepare a revision that addresses the issues, together with a cover letter explaining how you did so. My plan is to resend the revision to the present referees.

Reviewer 1 ·

Basic reporting

Basic reporting
Thank you for the submission of manuscript titled:
The evolution of SARS-CoV-2 and the COVID-19 pandemic (#83519). Here are my comments:

1. Paragraph 1 (Introduction) can be made simpler; then, update and decrease the references, especially those aged >10 years. I suggest to be more focused on coronavirus and the nearest pandemic before COVID-19 (Spanish Flu). Close sentence 1 with the newest pandemic, COVID-19.
2. Introduction: Add the estimate of viral genetic material error rate (X per one million base for example)  paragraph 1
3. Add long form of COVID-19, SARS-CoV-2 at the first occurrence
4. In addition to discussing Delta only, the authors should discuss about Omicron and cite it. Some references can be added: https://www.ncbi.nlm.nih.gov/pmc/articles/PMC9714996/, https://journals.lww.com/annals-of-medicine-and-surgery/Fulltext/2022/11000/Omicron_B_1_1_529_subvariant__Brief_evidence_and.28.aspx, https://www.sciencedirect.com/science/article/pii/S1319562X22002881.
5. For the last paragraph of the introduction, 1st to 3rd sentence implies that this is an original article, although it was stated as meta-analysis (however I am unsure what is meant by the authors since I did not find such article type as an appropriate method for this study) later on. I think these sentences can be deleted and then the authors can add about the study’s objective clearly and importance of this study afterwards.
METHODS
6. First sentence seems inaccurate, I encourage the authors to cite GISAID directly (https://gisaid.org/hcov19-variants/)
7. Cite only one reference for formula of mutation rate

Experimental design

This is quite confusing since I did not find what kind of meta-analysis done by the authors, it is seriously inconsistent with previous statement.

Validity of the findings

1. Figure 1 is more suitable on the result section, then discuss it
2. Divide figure 1 to figure 1a (continent) and 1b (country)
3. Why there are some discrepancies on total number of sequence (13,396,972 vs. 13,074,212)?
4. It is not acceptable to include citation in the result part  Overall result section is mainly incorrect in its current form, thus it must be totally rewritten
5. What is this? Several studies have suggested that the long-term evolutionary adaptation of SARS-CoV-2 with the limited genomic surveillance in replicating in host cells is the reason for this viral evolution. It is not allowed to discuss other studies’ findings on result section (as I mentioned, this is not a meta-analysis)
6. The authors must on discussing the result, not talking everything about other research
7. I also see the comparison with 2019, how about the explanation of this data?

Additional comments

1. Github file is not found
2. Add limitation and strength on discussion

Cite this review as

·

Basic reporting

na

Experimental design

na

Validity of the findings

na

Additional comments

The manuscript entitled “The evolution of SARS-CoV-2 and the COVID-19
Pandemic” covers an important topic of significant scientific relevance. The authors conducted a meta-analysis of 13,396,972 SARS-CoV-2 sequences obtained from GISAID by October 9th, 2022, and calculated the mutation rates of four structural proteins (N, M, S, and E) to assess their fitness in human hosts. They also conducted a correlation analysis and used machine learning tools to analyze the data. The figures were constructed using matplotlib and Seaborn, and the analysis was conducted in Pandas and Scikit-Learn. They detected the correlation of the mutation rates from each protein of SARS-CoV-2. In my opinion, this manuscript can be published after some correction/modification.

1. The Methods and Materials section is too short.
a) Provide more information about the GISAID sequences: It would be helpful to know how the 13,396,972 sequences were selected from GISAID, such as whether they were filtered for quality or specific geographic regions. Providing this information would help readers understand the representativeness of the dataset.
b) Provide more detail on the correlation analysis.
c) Explain the machine learning approach.
d) Authors should also mention the rational for choosing Pearson algorithm.
2. Results:
a) The result section also needs to be re written. The authors should describe the outcome of each plot/analysis. This section in its current form provides very limited and brief information.
b) The authors should extensively describe the figures and derived results. For example, authors only mention about correlation between S and M protein. Authors should also mention other correlations observed and later should discuss their scientific relevance in the discussion section in detail.
In short, authors should re write this section, explaining outcome of each plot/analysis and try to explain the outcome of each.
3. Discussion and Conclusion:
a) Clarify the key findings of the study and how they relate to the current understanding of SARS-CoV-2 evolution and its impact on public health.
b) Authors should also discuss other studies conducted related to the topic. Discus how the conclusion/finding of those studies support or contradict to the conclusion/finding of this study.
c) Provide more specific examples and evidence to support the claims made in the discussion section.
d) Highlight the novel or unique aspects of the study, particularly in relation to previous research on SARS-CoV-2 evolution and its impact on human health.
e) Discuss potential future directions for research in this area, including the need for additional studies to confirm and build upon the findings of this study.
f) Address any potential biases or limitations in the study design or data analysis that may affect the interpretation of the results.
g) Authors should also try to support the finding of this study with already published in vitro and in vivo studies exploring the mutation, pattern of mutation in different proteins. And effect of mutation in various protein and its effect on rate of infection.

---

## Round 0.2 · Minor Revisions

The authors have successfully addressed all the concerns raised during the review process through their revisions. However, the manuscript need an editing of the English. Use this example for an edited abstract to apply to the manuscript:

> Scientists have made great efforts to understand the evolution of SARS-CoV-2 (Severe Acute Respiratory Syndrome Coronavirus 2) to provide information to public health experts on strategies to control this viral pathogen. The pandemic of the coronavirus disease that began in 2019, COVID-19, lasted nearly three years, and nearly all countries have set different epidemic prevention policies for this virus. The continuous evolution of SARS-CoV-2 alters its pathogenicity and infectivity in human hosts, thus the policy and treatments have been continually adjusted. Based on our previous study on the dynamics of binding ability prediction between the COVID-19 spike protein and human ACE2, the present study mined over 10 million sequences and epidemiological data of SARS-CoV-2 during 2020-2022 to understand the evolutionary path of SARS-CoV-2. We analyzed and predicted the mutation rates of the whole genome and main proteins of SARS-CoV-2 from different populations to understand the adaptive relationship between humans and COVID-19. The study identified a correlation of the mutation rates from each protein of SARS-CoV-2 and various human populations. Overall, this meta-analysis provides a scientific basis for developing data-driven strategies to confront human pathogens.

Pay attention to spaces preceding the opening of parentheses.

Please explain "Pango counts", it wasn't clear from the methods or results.

Be sure to correct all typos of "SRAS-CoV-2".

Reviewer 1 ·

Basic reporting

1. The authors did not do my suggestion to reduce references (now it is still >45 references, just in the introduction). Please remove it again, especially in the first and second paragraphs.
2. Mutation rate is wrong, write 10^4, 10^5, etc (I mean without negative sign, not necessary to include ^)
3. Make long form of RNA, mrNA, DNA

Experimental design

1. I am still unsure about meta-analysis and still encourage to remove it. From the below example ( https://www.frontiersin.org/articles/10.3389/fnins.2020.00209/full), meta-analysis was still conducted by risk of bias assessment, etc. Thus, I recommend it to make as a bioinformatic analysis or similar term. Last paragraph of introduction. If the authors can give an appropriate citation (or give a good example) about their correct design from the prior research, maybe acceptance can be given. However, for now, as far as I know, meta-analysis is really different from this study (also applies for writing in the methodology), and it must be clarified further.

Validity of the findings

The result is interesting. However, I am still unsure about the use of meta-analysis (which as far as I know is related with pooled data from published literature). This concern must be adequately addressed before further evaluation process.

Additional comments

Check for typographic error:
1. includde (result section)
2. Rewrite: Several studies have suggested that the long-term evolutionary adaptation of SARS-CoV-2 with the limited genomic surveillance in replicating in host cells is the reason for this viral evolution --> Several studies have suggested that the long-term evolutionary adaptation of SARS-CoV-2 and
limited genomic surveillance are the reasons for this viral evolution

Cite this review as

·

Basic reporting

na

Experimental design

na

Validity of the findings

na

Additional comments

I have carefully reviewed the manuscript and am happy to let you know that it can be accepted in its present form for publication in this journal. The revisions made by the authors have addressed all the concerns raised during the review process and have greatly improved the quality of the manuscript. The research presented is rigorous, the methodology is sound, and the results are significant and relevant to the field. The manuscript is well-written and organized, making it suitable for publication in this journal. I commend the authors for their efforts in incorporating the suggested changes and believe this work will make a valuable contribution to the scientific community.

---

## Round 0.3 · accepted · Accept

Authors have improved English language but there is still scope for some improvement - especially in the first sentence.